# Cholesterol-stabilized membrane-active nanopores with anticancer activities

Jie Shen[1,2], Yongting Gu[1], Lingjie Ke[1], Qiuping Zhang[1], Yin Cao[1], Yuchao Lin[1], Zhen Wu [1], Caisheng Wu[1], Yuguang Mu [3], Yun-Long Wu [1], Changliang Ren [1,4] ✉ & Huaqiang Zeng [2] ✉

Cholesterol-enhanced pore formation is one evolutionary means cholesterol-free bacterial cells utilize to specifically target cholesterol-rich eukaryotic cells, thus escaping the toxicity these membrane-lytic pores might have brought onto themselves. Here, we present a class of artificial cholesterol-dependent nanopores, manifesting nanopore formation sensitivity, up-regulated by cholesterol of up to 50 mol% (relative to the lipid molecules). The high modularity in the amphiphilic molecular backbone enables a facile tuning of pore size and consequently channel activity. Possessing a nano-sized cavity of ~1.6 nm in diameter, our most active channel **Ch-C1** can transport nanometer-sized molecules as large as 5(6)-carboxyfluorescein and display potent anticancer activity ($IC_{50}$ = 3.8 μM) toward human hepatocellular carcinomas, with high selectivity index values of 12.5 and >130 against normal human liver and kidney cells, respectively.

As an essential structural component of eukaryotic cell membranes[1,2], cholesterol is typically absent from bacterial cell membranes[3]. By leveraging on this difference, bacterial cells manage to evolve cholesterol-dependent cytolysins (CDCs), a family of bacterial protein toxins that form circular membrane-lytic pores only in the presence of cholesterol to specifically target and harm eukaryotic cells, but not themselves[4]. Other protein channels, which display similarly interesting ion transport activities up-regulated by cholesterol, include nicotinic acetylcholine receptor[5], epithelial Na+ channels[6], transient receptor potential channels[7] and Alzheimer β-amyloid peptides[8]. Nevertheless, despite a number of examples seen in Nature, within the realm of artificially created membrane transporters[9–80], presence of cholesterol is often observed to cause several fold decreases in ion transport activity, and cholesterol-induced enhancement in ion transport activity, to the best of our knowledge, has yet to be demonstrated.

In this work, we report on the examples of cholic acid-derived pore-formers, forming multimeric nanopores that become increasingly active in ion transport when the level of cholesterol increases from 0 to 50 mol% relative to the amount of lipid molecules present in the membrane. Among this series of structurally similar pore-formers, the most active ensemble formed by **Ch-C1** encloses a nano-sized cavity of 1.6 nm in diameter, which can be modularly tuned down through side chain modification. Importantly, **Ch-C1** further displays potent anticancer activity ($IC_{50}$ = 3.8 μM) toward human liver cancer cells with low cytotoxicity to normal human liver ($IC_{50}$ = 47.5 μM) and kidney cells ($IC_{50}$ > 500 μM).

## Results and Discussion

### Molecular design of pore-forming Ch-Pns

Attracting wide interests from biomimetic chemists, artificial trans-membrane pores can be readily constructed using backbone-rigidified foldamers[48,67,78–80]. We however envisioned that this type of pores with high structural rigidity unlikely will exhibit cholesterol-enhanced ion transport activities. Instead, those pore-enclosing conformations, which are generated from either a single molecular backbone without a defined conformation[9,20,28,39,65] or multiple components assembled via non-covalent forces, are deemed to be more sensitive toward environmental stimuli. In this regard, there exist a wide range of

[1]Fujian Provincial Key Laboratory of Innovative Drug Target Research and State Key Laboratory of Cellular Stress Biology, School of Pharmaceutical Sciences, Xiamen University, Xiamen, Fujian 361102, China. [2]College of Chemistry, Fuzhou University, Fuzhou, Fujian 350116, China. [3]School of Biological Sciences, Nanyang Technological University, 60 Nanyang Drive 637551, Singapore. [4]Shenzhen Research Institute of Xiamen University, Shenzhen, Guangdong 518057, China. ✉e-mail: changliang.ren@xmu.edu.cn; hqzeng@fzu.edu.cn

strategies in the literature. Inspiring ones include helical folding via solvophobic forces by Moore[9], "rigid rod" β-barrels by Matile[13–15], tail-to-tail assembly of half channel molecules[16] and pore-forming steroid-modified biscarbamate[17] by Kobuke, dendritic[19] or linear-shaped steroid derivatives[19] by Regen, columnar stacking involving shape-persistent macrocycles by Gong[23,27,49], multiblock amphiphiles having alternatively arranged hydrophilic and hydrophobic units by Muraoka and Kinbara[28,39,65], structurally simple pore-forming mono- and tripeptides[71,72] and trimesic amides[73] by Zeng, as well as H-bond-assisted formation of pore-containing rosettes derived from mannitol[40] or diol-containing 1,3-diethynylbenzene[52] by Talukdar, folate by Matile[18], G-quartets by Davis[22] and Dash[44,66] and a fused guanine–cytosine base by Wanunu[69].

With long-standing interests in developing novel membrane-active artificial transporters[71–81], we recently reported molecular machine-inspired molecular swings as a highly efficient K[+] transporter[82]. These molecular swings are constructed by attaching a cation-binding and -transporting crown ether onto the center of a rigid cholic acid-derived linear scaffold via a flexible linker. Possessing two cholic acid units at the ends, this rigid cholic acid-derived scaffold is designed to have a length of 40 Å in for spanning the hydrophobic lipid region (Fig. 1a, b). Inspired by the unusual ability of Amphotericin B (AmB) to form an ergosterol-assisted barrel stave-like transmembrane pore, we wonder correspondingly whether the cholic acid-derived linear scaffold would be able to assemble into a form of circular nanopore in a cholesterol-dependent manner (Fig. 1c). To examine this possibility, we initially tested a series of four structurally similar molecules **Ch-Pn**s ($n$ = 1, 3, 5 and 7, Fig. 1a) that contain one to seven ethylene glycol units, and one control molecule **Ch-C1** that has just a methyl group in its side chain (Fig. 1a, b).

### Ch-Pns exhibit cholesterol-enhanced ion transport activities

The ability of these five molecules (**Ch-Pn**s and **Ch-C1**) to self-assemble into a membrane-active ion-transporting pore in the

presence of cholesterol (**Ch**) molecules was assessed using the well-established pH-sensitive fluorescence assay (Fig. 2a), employing large unilamellar vesicles (LUVs) containing EYPC (egg yolk phosphatidylcholine) and **Ch** at varying molar ratios of 1:0 to 1:1. In typical conditions, pH-sensitive HPTS (8-hydroxy-1,3,6-pyrenetrisulfonate, 0.1 mM) and NaCl (100 mM) were entrapped in EYPC-based LUVs containing 0 mol%, 25 mol%, 50 mol%, 75 mol% and 100 mol% of **Ch** with respect to lipids in HEPES buffer at pH 7. The LUV stock solution was then diluted to the same buffer at pH 8.0 to create a pH gradient of 1.0 across the membrane. Upon addition of channels at a final concentration of 8 μM (3.7 mol% relative to lipid), the changes in ratiometric value of the fluorescence intensity at 510 nm ($I_{460}/I_{403}$) of the encapsulated HPTS were recorded for 300 s. After subtracting the background intensity at t = 300 s in the absence of the channel molecules (Supplementary Fig. 2), the signals were further normalized to give fractional ion transport activities summarized in Fig. 2b (For detailed ion transport curves, see: Supplementary Figs. 3, 4). It is clear that, in the absence of **Ch**, all five channels display weak ion transport activities of ≤16%. Nevertheless, increasing membrane **Ch** content from 0 mol% to 50 mol% makes these channels increasingly more active, generating 4–6 folds enhancements in ion transport activity at 50 mol% of **Ch** for all five channels. This interestingly general trend highlights a pivotal role that **Ch** molecules play in augmenting self-assembling propensities of these molecules. Coincidently, this 50 mol% is also the highest **Ch** content possible in eukaryotic cells. Further increases in **Ch** content to 100 mol% result in poorer performances, likely as a result of **Ch**-mediated over-stiffness of the membrane that attenuates the pore-forming capacity of these molecules[83].

### Ch-Pns are less active than Ch-Cns ($n \neq 8$)

To our surprise, the control molecule **Ch-C1**, having no ethylene glycol units, consistently exhibits better ion transport activities than **Ch-Pn**s at all **Ch** contents. Although we are not exactly sure about the reasons

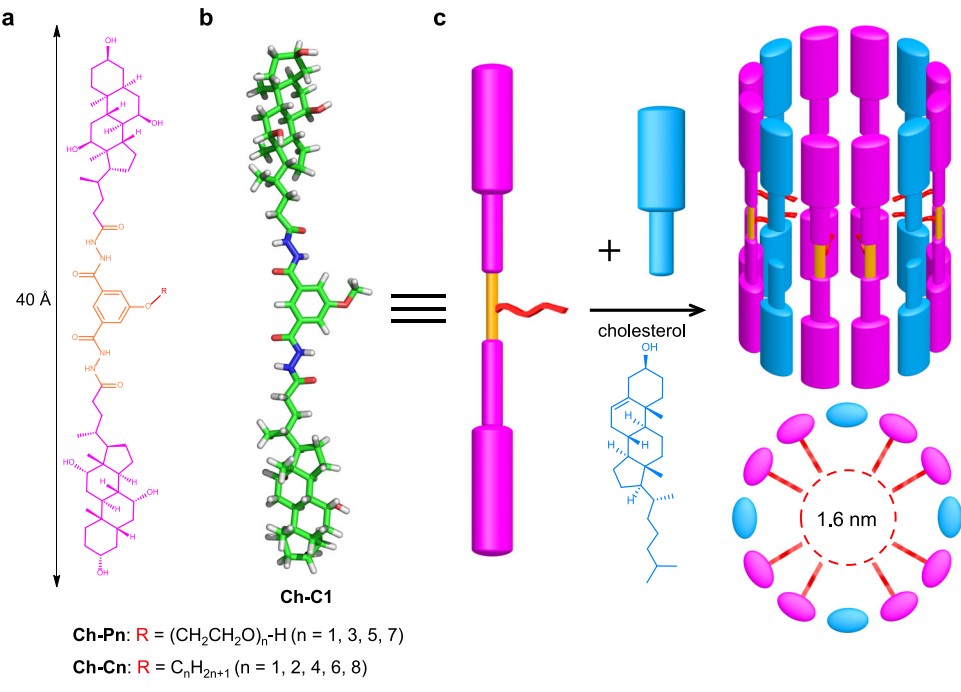

**Ch-Pn**: R = (CH₂CH₂O)ₙ-H (n = 1, 3, 5, 7)

**Ch-Cn**: R = CₙH₂ₙ₊₁ (n = 1, 2, 4, 6, 8)

**Ch-H**: R = H

**Fig. 1 | Molecular design and self-assembly of nanopores. a** Chemical structures of cholic acid-derived amphiphilic transmembrane channels **Ch-Pn**, **Ch-Cn** and **Ch-H**. **b** Computationally optimized structure of **Ch-C1** at the level of B3LYP/6-31 G* in the gas phase. **c** Schematic illustration of one possible means to

form a toroidal channel of ~1.6 nm in diameter by **Ch-C1**, a process that is primarily assisted by cholesterol molecules (**Ch** in blue), for mediating ion transport across the membrane.

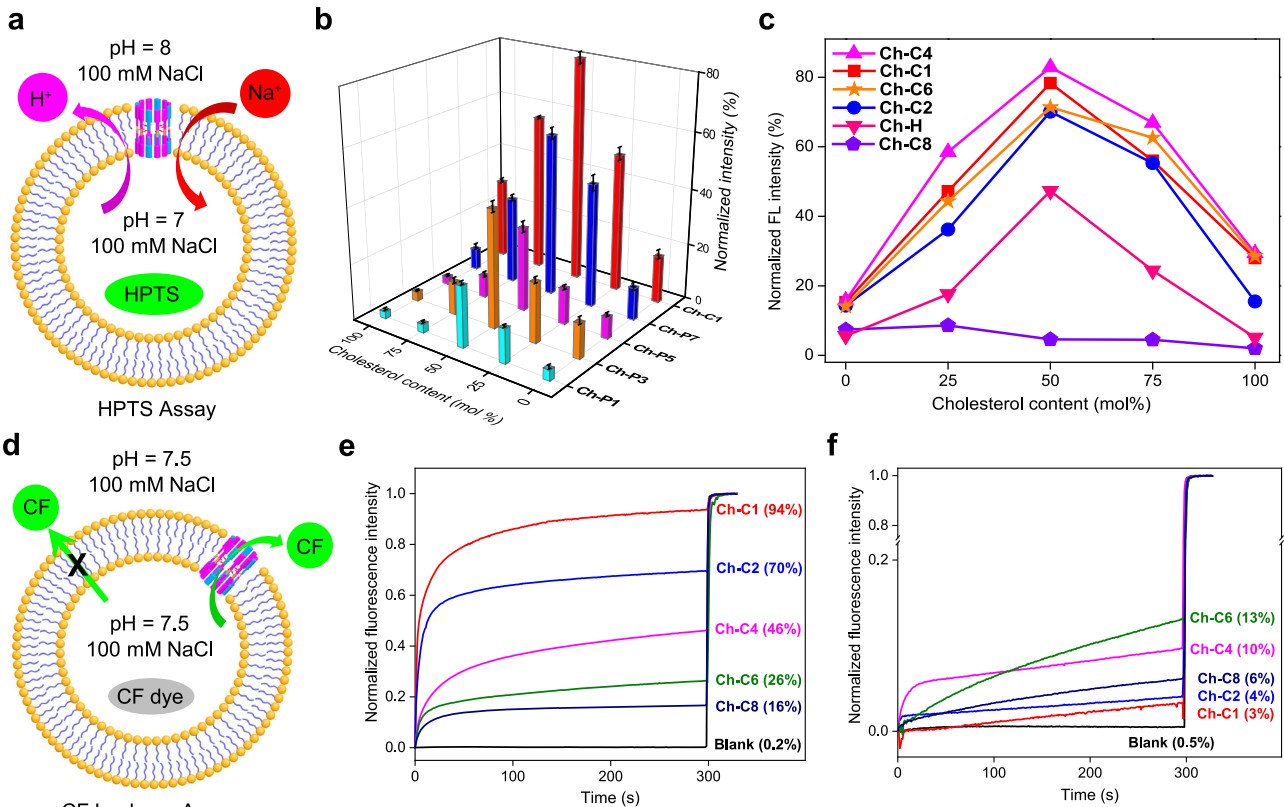

**Fig. 2 | Transmembrane transport activity study of nanopores. a** Schematic illustration of HPTS assay. HPTS = 8-hydroxy-1,3,6-pyrenetrisulfonate. **b, c** Normalized ion transport activities of channels over a duration of 5 min at 8 μM using LUVs containing different contents of cholesterol (0, 25, 50, 75 and 100 mol % relative to lipid) in the HPTS assay. **d** Schematic illustration of CF dye leakage assay for qualitatively probing pore size formed by channels **Ch-C1, Ch-C2, Ch-C4, Ch-C6** and **Ch-C8**. CF = 5(6)-carboxyfluorescein. **e** and **f** describe changes in fluorescence intensity of self-quenching CF dye ($\lambda_{ex}$ = 492 nm, $\lambda_{em}$ = 517 nm) after additions of channel molecules at 8 μM in the **e** presence or **f** absence of 50 mol % cholesterol relative to lipid. In **b** and **c**, all the relative errors are less than 3% based on a triplicate run. These experiments demonstrate that (i) cholesterol molecules do assist the formation of pores of >1 nm and (ii) side chain elongation decreases the pore size, suggesting side chains to orient toward, not away from the pore interior. Source data are provided as a Source Data file.

behind, this somewhat intriguing finding does suggest oxygen-rich ethylene glycol chains not to be that essential in order to facilitate transmembrane ion flux.

Logically, in the next step, we therefore decided to look into pore-forming capacity of four more molecules **Ch-Cn**s ($n$ = 2, 4, 6 and 8, Fig. 1a), which contain a straight hydrocarbon chain, as well as another control molecule **Ch-H**, which possesses a hydroxyl group. Similarly, the ion transport activities of these five channels are sensitively and increasingly up-regulated by increasing **Ch** contents of up to 50 mol% beyond which activities drop (Fig. 2c and Supplementary Fig. 5). Except for **Ch-C8**, the other four alkyl-containing **Ch-Cn**s ($n$ = 1, 2, 4 and 6) are more active than ethylene glycol-containing **Ch-Pn**s ($n$ = 1, 3, 5 and 7). In more details, at 50 mol% of **Ch**, **Ch-Cn**s ($n$ = 1, 2, 4 and 6) display fractional ion transport activities of 78%, 70%, 83% and 71%, but **Ch-Pn**s ($n$ = 1, 3, 5 and 7) show much weaker activities of 18%, 40%, 28% and 55%, respectively.

### Ch-Cns generate cholesterol-stabilized pores in the cholesterol-rich lipid bilayer

Since these linear-shaped molecules **Ch-Cn**s or **Ch-Pn**s carry no discernable binding groups for binding and transporting ions, we proposed a toroidal model where multiple linear molecules are circularly arranged to produce a multimeric hollow ensemble sensitive to cholesterol molecules and possibly lipid molecules as well (Fig. 1c). The stability, pore size and ion-transporting potential of the resultant nanopores are critically dependent on the **Ch** content as well as the length and orientation of side chains. In terms of side-chain

orientation, they may point toward the channel's interior as illustrated in Fig. 1c, or away from the interior. The former is believed to be a more likely scenario that is consistent with a negligible ion transport activity of 4.6% exhibited by **Ch-C8** (Fig. 2c), especially when compared to high activities of >70% by those within the same series but with side chains shortened by 2–7 carbon atoms (e.g., **Ch-Cn**s, $n$ = 1, 2, 4 and 6). In other words, we hypothesized that it is only when the side chains point toward the interior that a sharp decline in activity from 71% for hexyl-containing **Ch-C6** to 4.6% for octyl-containing **Ch-C8** may become possible.

To lend some support to this hypothesis, we applied self-quenching 5(6)-carboxyfluorescein (CF) dye leakage assay (Fig. 2d) to qualitatively evaluate the pore sizes and activities of **Ch-Cn**s ($n$ = 1, 2, 4, 6 and 8). Here, CF dye is highly fluorescent at diluted concentrations but self-quenches at high concentration of 50 mM via a self-dimerization process. Having the smallest dimension of 1.0 nm x 1.0 nm (Supplementary Fig. 6), CF molecules trapped inside LUVs can only permeate through a pore of >1 nm to reach the extravesicular region, resulting in increased fluorescence intensity. For this CF leakage assay, we prepared LUVs containing 50 mol% of **Ch** relative to lipids. As summarized in Fig. 2e, **Ch-C1** at 8 μM (3.7 mol% relative to lipid) elicits CF efflux by 94%, which is in sharp contrast to 3% CF efflux obtained using LUVs free of **Ch** (Fig. 2f). This high activity, which confirms the pore size formed by **Ch-C1** to be >1 nm, is followed by **Ch-C2** (70%), **Ch-C4** (46%), **Ch-C6** (26%) and **Ch-C8** (16%), indicating increasingly shrunk pore sizes by increasingly lengthened side chains among this series of channels. As summarized in Table 1 and

Supplementary Figs. 7, 8, the corresponding $EC_{50}$ values in the presence of **Ch** are 5.0, 7.0 and 7.9 μM for **Ch-C1**, **Ch-C2** and **Ch-C4**, respectively, and greater than 20 μM for both **Ch-C6** and **Ch-C8**. For comparison, the $EC_{50}$ value for melittin, a pore-forming toxin with estimated pore sizes of 3.5–4.5 nm[84], was 0.17 μM (0.08 mol% relative to lipid). In addition, none of these channels is capable of efficient transport of CF dye in the **Ch**-free membrane environment (Fig. 2f and Supplementary Fig. 9).

Taken together, the above CF dye leakage data are in great accord with the AmB-inspired hypothetic barrel stave-like model, i.e., **Ch** molecules may aid the formation of a nanoscale toroidal pore by alternatively interacting with channel molecules whose side chains point toward the pore interior (Fig. 1c). Forming these large pores is also consistent with the ability of **Ch-C1** to transport anions (Supplementary Fig. 10), and with the fact that these large pores makes the HPTS assay insignificantly influenced by ions and/or HPTS dye molecules that get transported into the extravesicular region (Supplementary Figs. 11, 12) and thus do not generate a well-defined activity order among **Ch-Cn**s (Fig. 2c).

**Table 1 | EC₅₀ values (μM) determined using CF-leakage assay**

|  | EC₅₀ (μM) | n |
| --- | --- | --- |
| **Ch-C1** | 5.0 ± 0.2 | 4.08 ± 0.60 |
| **Ch-C2** | 7.0 ± 0.2 | 6.40 ± 1.02 |
| **Ch-C4** | 7.9 ± 0.4 | 3.58 ± 0.62 |
| **Ch-C6** | >20 | - |
| **Ch-C8** | >20 | - |
| **Ch-H** | 8.0 ± 0.1 | 6.39 ± 0.37 |
| **Melittin** | 0.17 ± 0.01 | 1.33 ± 0.07 |

## Ch-Cns generate toroidal pores of <2 nm in diameter

To quantify the pore sizes and also verify the single channel behaviors of the pore-formers, we conducted planar lipid bilayer experiments using the bilayer membrane made up of 1,2-diphytanoyl-sn-glycero-3-phosphocholine and **Ch** in a 2:1 molar ratio in symmetric (*cis* chamber = *trans* chamber = 1 M KCl) baths. The recorded single channel current traces from –100 mV to 100 mV for **Ch-C1** undoubtedly confirm its ability to transport ions through a channel mechanism (Fig. 3a). At a lower voltage of −20 mV, channels appear to exist in the form of barrel-shaped ensemble mostly having a single diameter. At other voltages, many multiple level transitions were observed in the current traces, suggesting the existence of toroidal pores of varied diameters[85,86]. These varied diameters result from dynamic breathing-type interactions of **Ch-C1** with **Ch** and these interactions might be further influences by the lipid molecules that are also in constant movement.

Accordingly, the histogram of currents at all voltages were plotted to obtain the mean current values and the relative errors (Supplementary Figs. 13–15) for plotting the current-voltage (I-V) curve presented in Fig. 3a. From the linear I-V curve, the single-channel conductance was calculated to be 534.7 ± 18.1 pS, corresponding to a highly efficient transport of $3.4 \times 10^8$ ions/s at 100 mV. On the basis of this conductance value, the channel's pore size was estimated to be 1.67 nm across by using the Hille equation[87] (Eq. 4). In contrast, in the single channel current measurements using **Ch**-free bilayer membrane, we didn't observe any channel activity even after prolonged recordings, confirming the critical role of **Ch** molecules in the pore formation. Repeating the single channel current recording twice followed by pore size calculations (Supplementary Figs. 16–20) gives an average conductance value of 496.4 ± 46.5 pS and an average pore size of 1.61 ± 0.09 nm.

The same set of single channel conductance measurement and pore size calculation performed on **Ch-C4** in triplicate gives an average

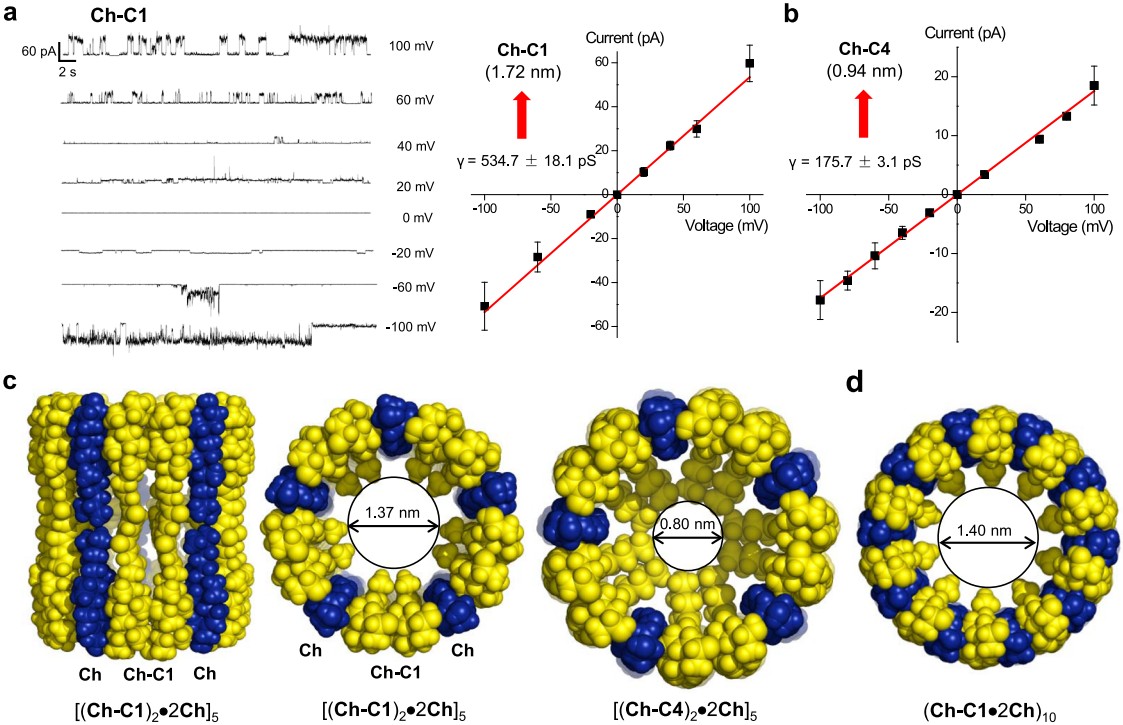

**Fig. 3 | Single channel study and molecular dynamic simulation of nanopores. a** Single channel current traces of **Ch-C1** recorded in symmetric (*cis* chamber = *trans* chamber = 1 M KCl) baths, the red lines refer to the mean current values for plotting current-voltage (I-V) curve for obtaining the ion conductance (γ) and pore size (1.72 nm) for **Ch-C1**. **b** I-V curve and estimated pore size (0.94 nm) for **Ch-C4**. Data are represented as mean current values ± half width at half maximum. *n* = 3 independent experiments. **c** Molecular dynamics-simulated pore-forming ensembles of different sizes made up of channel (**Ch-C1** or **Ch-C4** in yellow) and **Ch** (in blue) molecules in 1:1 molar ratio. **d** An alternative self-assembling possibility involving **Ch-C1** and **Ch** in 1:2 molar ratio to generate a pore of 1.40 nm.

conductance value of $188.5 \pm 9.6$ pS (Supplementary Figs. 21–28) and an average pore size of $0.94 \pm 0.03$ nm in diameter. A representative I-V curve presented in Fig. 3b gives a conductance value of $175.7 \pm 3.1$ pS and a pore size of 0.90 nm in diameter.

## Pore-formers do associate with cholesterol molecules

The binding between the channel molecules and **Ch** was initially supported by $^1H$ NMR titration experiments involving titrating 0–100 equivalents of **Ch** into a THF-$d_8$ solution containing **Ch-C2** at 5 mM (Supplementary Fig. 29). Upon addition of up to 100 equivalents of **Ch**, the signals corresponding to the two amide protons of **Ch-C2** show obvious downfield shifts of up to 0.08 and 0.16 ppm, respectively, indicating that not only **Ch-C2** can bind to **Ch** molecules but also such binding likely increases the self-association extent of **Ch-C2** molecules. Using UV-vis spectroscopy, we observed (1) red shifts of 2.5 nm and 21 nm for the maximum and minor absorption peaks when increasing the concentration of **Ch-C2** from $5\,\mu M$ to $80\,\mu M$ in THF at 20 °C (Supplementary Note 1 and Supplementary Fig. 30) and (2) a red shift of 2.5 nm and a 24% decrease in intensity for the maximum absorption peak (Supplementary Fig. 31). These two pieces of data are evidently suggestive of associations among **Ch-C2** molecules and between **Ch-C2** and **Ch** molecules. Consistent with the $^1H$ NMR- and UV-vis-based binding data, analysis of the high-resolution mass spectra (HRMS) of the THF solution containing **Ch-C2** ($10\,\mu M$) and **Ch** (1 mM) reveals dimeric $(\mathbf{Ch\text{-}C2})_2 \bullet Na^+$ and $(\mathbf{Ch\text{-}C2})_2 \bullet H^+$ as well as $4\mathbf{Ch\text{-}C2} \bullet 2\mathbf{Ch} \bullet 2H^+$ as the major peaks (Supplementary Figs. 33, 34), followed by much weaker signals, corresponding to the singly charged $(\mathbf{Ch\text{-}C2})_2 \bullet H^+$ that associates with one or two **Ch** molecules (Supplementary Figs. 35, 36). These HRMS data suggest self-dimerization involving two **Ch-C2** molecules to be far much stronger than the mutual-association between **Ch-C2** and **Ch** molecules.

## Computational models of the pores formed by Ch-C1 and Ch-C4

In light of (1) the side chain-dependent activity trend deduced from the CF leakage assay, (2) the quantified pore sizes of 1.61 and 0.94 nm for **Ch-C1** and **Ch-C4**, respectively, and (3) the HRMS data that confirm the existence of a dimeric $(\mathbf{Ch\text{-}C1})_2$ fragment, we constructed a few **Ch**-containing toroidal ensembles $[(\mathbf{Ch\text{-}Cn})_2 \bullet 2\mathbf{Ch})_m]$, with side chains pointing toward the pore interior (Fig. 3c and Supplementary Fig. 37a). This was followed by molecular dynamics simulation (MD) to yield pore sizes of 1.37 and 0.80 nm for $[(\mathbf{Ch\text{-}C1})_2 \bullet 2\mathbf{Ch})_5]$ and $[(\mathbf{Ch\text{-}C4})_2 \bullet 2\mathbf{Ch})_5]$, respectively. Moreover, ensembles $[(\mathbf{Ch\text{-}Cn})_2 \bullet 2\mathbf{Ch})_m]$ with $m \neq 5$ generate a pore size that is too small or too large (Supplementary Fig. 37a), and those with half of side chains pointing outward are energetically less stable (Supplementary Fig. 37b). We thus believe pentameric pores containing inward-pointing side chains might be one preferred association mode at least for **Ch-C1** and **Ch-C4**, enabling channel and **Ch** molecules to act synergistically to produce membrane-active wide pore ensembles. Certainly, there might exist other possible structural models that may also enclose a pore size comparable to the experimentally determined one. Figure 3d illustrates one such ensemble $(\mathbf{Ch\text{-}C1} \bullet 2\mathbf{Ch})_{10}$, having a pore size of 1.40 nm that closely matches the pore size of ~1.6 nm for **Ch-C1**.

## Ch-C1 mediates potent and specific anticancer effects

Given that disrupting ion homeostasis across the cellular membrane could result in cell death[31,52,88–90], it is expected that these hole-punching channels might exert toxicity on cancer cells. In this regard, we applied a standard cell viability assay to evaluate the anticancer activities of **Ch-Cn**s against human hepatocellular carcinomas (HepG2 cell line, Table 2 and Supplementary Fig. 38). We found that channels with different side chain lengths or pore sizes show good anticancer activities. Among them, the most potent **Ch-C1** exhibits an $IC_{50}$ value of as low as $3.8\,\mu M$, which is comparable to those of the well-known chemotherapeutic agents doxorubicin ($1.5\,\mu M$) and paxlitaxel ($8.2\,\mu M$)

**Table 2 | $IC_{50}$ values (μM) for Ch-Cn (n = 1 – 8) against HepG2 and U87-MG cancer cells**

| | $IC_{50}$ (μM) | |
|---|---|---|
| | **HepG2** | **U87-MG** |
| **Ch-C1** | 3.8 | 50.7 |
| **Ch-C2** | 8.4 | 186.7 |
| **Ch-C4** | 26.6 | > 500 |
| **Ch-C6** | 36.6 | > 500 |
| **Ch-C8** | 57.5 | > 500 |
| **Cisplatin** | >500 | > 500 |
| **Paclitaxel** | 8.2 | > 500 |
| **Doxorubicin** | 1.5 | > 500 |

and much lower than that of cisplatin (>500 μM) (Table 2 and Supplementary Fig. 38). For human primary glioblastoma cell line (U87-MG) that are refractory to be treated, the above-mentioned three chemotherapeutic agents (cisplatin, doxorubicin and paclitaxel) show no activity ($IC_{50} > 500\,\mu M$, Table 2 and Supplementary Fig. 39), whereas **Ch-C1** still displays significant therapeutic activity ($IC_{50} = 50.7\,\mu M$) (Table 2 and Supplementary Fig. 39). Not very surprisingly, for both cancer cell lines, the anticancer activities of these channels correlate well with the pore sizes (or the side chain lengths), i.e., larger pores (or shorter side chains) show more potent activities.

These good anticancer activities of **Ch-C1** promoted us to further evaluate its selectivity and safety towards normal cells by performing the cell viability assay on human normal liver cells (THLE-2) and human renal proximal tubular epithelial cells (HK-2). Desirably, the determined $IC_{50}$ values of **Ch-C1** against THLE-2 cells and HK-2 cells were 47.5 and >500 μM, respectively (Supplementary Fig. 40). Defined as the ratio of $IC_{50}$ values between normal and cancer cells, the selectivity indexes (SI) determined for **Ch-C1** molecules are 12.5 and >130 against THLE-2 and HK-2 cells, respectively. These high SI values are more than one order of magnitude higher than those of doxorubicin and paclitaxel[91,92], indicating comparably lower cytotoxicities of **Ch-C1** to normal cells than doxorubicin and paclitaxel. We then employed mice blood cells to test the in vitro hemolytic activity of **Ch-C1**. To our delight, **Ch-C1** also showed a low hemolytic activity. The concentration causing 50% hemolysis of red blood cells ($HC_{50}$) is 127.3 μM (Supplementary Fig. 41), corresponding to a high SI value of 38.5. This high specificity may be caused by the high-level expression of cholic acid receptors on hepatocarcinoma cells that leads to higher uptake of cholic acid-containing channel molecules[93,94].

Two types of dye molecules (PI and DAPI) were applied to assess the membrane integrity in the presence of **Ch-C1** molecules. While DAPI molecules can enter the cells at high concentrations and become blue upon binding to the AT regions of dsDNA, membrane-impermeable PI enters cells having compromised membranes and binds tightly to the intracellular nucleic acids to emit red fluorescence when excited at 535 nm. After the incubation of HepG2 cells with various concentrations of **Ch-C1** (0, 1, 4 and 16 μM) for 36 h, the cells were fixed, stained with PI and DAPI, and analyzed under a laser confocal microscope. As can be seen from Fig. 4, the DAPI-stained cells reveal intact cell membrane, confirming that pore-forming **Ch-C1** does not disrupt the cell membranes to a noticeable extent at concentrations of up to 16 μM. And the PI-stained cells do suggest that **Ch-C1** make the membrane leaky and more permeable via forming wide pores.

It has been shown that disruption of cellular homeostasis caused by leaky cell membrane leads to cell apoptosis via the caspase signaling pathway[95–97]. Applying dead cell apoptosis kit, we treated the HepG2 cells with **Ch-C1** at 16 μM for 24 h, stained the cells using both green Annexin V- FITC conjugate and red PI dyes and sorted the cells by flow cytometry. Considering that cells stainable by Annexin V-FITC

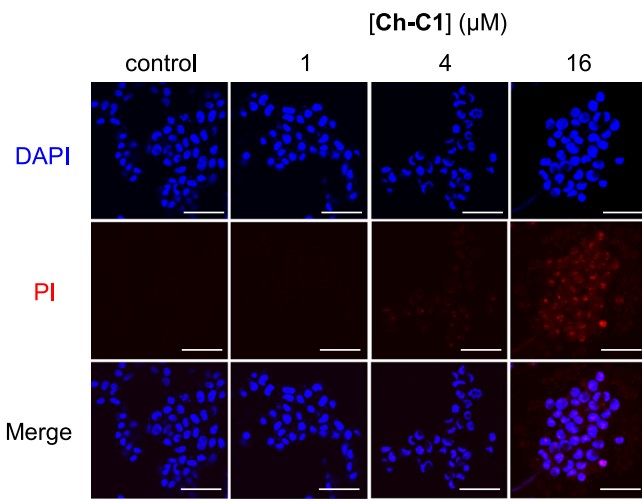

**[Ch-C1] (μM)**

control     1     4     16

DAPI

PI

Merge

**Fig. 4 | Nanopores punch holes on cancer cells.** Cell imaging of HepG2 cells treated with **Ch-C1** at concentrations of 1 μM, 4 μM and 16 μM for 36 h, followed by staining with blue DAPI and red PI dyes. Blue and red images were merged by ImageJ 1.8.0. Each experiment was repeated twice independently with similar results. Scale bar: 50 μm. DAPI 4′,6-diamidino-2-phenylindole, PI propidium iodide.

conjugate correspond to early or later apoptotic cells (Fig. 5a), the fact that the percentage of apoptotic cells substantially increases from 1.07% to 8.88% with increased concentrations of **Ch-C1** from 0 to 16 μM establishes the capability of **Ch-C1** to induce cell apoptosis. Some characteristic proteins involved in apoptosis include (1) apoptosis initiator protein (caspase-9) that initiates cell killing but is cleaved during early apoptosis[98,99], (2) poly(ADP-ribose) polymerase (PARP) cleaved by the activated caspase 9 to facilitate apoptosis by preventing DNA repair and (3) anti-apoptotic protein (Bcl-2) that undergoes apoptosis- induced inhibition. Therefore, following an apoptotic stimulus, decreased presences of caspase 9, PARP and Bcl-2, together with the increased presence of cleaved PARP, are expected. And these are indeed what we observed when we treated HepG2 cells with **Ch-C1** at 16 μM for 6 h (Fig. 5b, c), prompting us to conclude that **Ch-C1**-enhanced membrane permeability can induce HepG2 cell apoptosis via the caspase 9 pathway.

In conclusion, we have conceptualized and demonstrated a class of cholesterol-enhanced nanopores in analogy to the barrel stave-like pores formed by AmB in the presence of ergosterol in the membrane. In assisting the wide pore formation, two cholesterol molecules, which may adopt a bilayer-type structure, alternatively interact with the self-dimerized channel molecules in a 1:1 molar ratio in a concentration-dependent manner, with the most active (likely also the most stable) pores always formed when cholesterol is at 50 mol % with respective to lipid molecules. The pores thus formed further show side chain-dependent activities, and the one with the shortest methyl side chains (e.g., **Ch-C1**) produces the widest pore of about 1.6 nm across and the highest cross-membrane transport activity. Channel **Ch-C1** further displays the best anti-hepatic cancer activity and high specificity to tumor cells, and may serve as a promising candidate for hepatic cancer treatments. This artificial cholesterol-enhanced nanopore system with high structural modularity offers a facile strategy to construct a family of structurally diversified sensitive nanopores, possibly giving rise to a broad range of applications including cholesterol sensors, anticancer agents and controlled drug delivery.

## Methods

### Ion transport study using the HPTS assay
Egg yolk L-α-phosphatidylcholine (EYPC, 0.6 ml, 25 mg/mL in CHCl₃, Avanti Polar Lipids, USA) and cholesterol (0, 1.9, 3.8, 5.7 or 7.6 mg) were dissolved in CHCl₃ (10 mL). CHCl₃ was removed under reduced

pressure at 35 °C. After drying the resulting film under high vacuum overnight at room temperature, the film was hydrated with 1.5 mL 10 mM HEPES buffer solution (100 mM NaCl, pH = 7.0) containing a pH sensitive HPTS dye (0.1 mM) at 37 °C for 2 h to give a milky suspension. The mixture was then subjected to 8 freeze-thaw cycles. The vesicle suspension was extruded through polycarbonate membrane (0.1 μm) to produce a homogeneous suspension of LUVs. The suspension of LUVs was dialyzed for 16 h with gentle stirring (300 r/min, 4 °C) using membrane tube (MWCO = 10,000 Da) against the same HEPES buffer solution (300 mL, without HPTS) for 6 times to remove the unencapsulated HPTS to yield LUVs with lipids at a concentration of 13 mM.

The HPTS-containing LUV suspension (30 μL, 13 mM in 10 mM HEPES buffer containing 100 mM NaCl at pH = 7.0) was added to a HEPES buffer solution (1.75 mL, 10 mM HEPES, 100 mM NaCl at pH = 8.0) to create a pH gradient for ion transport study. A solution of channels in DMSO was then injected into the suspension under gentle stirring. Upon the addition of channels, the emission of HPTS was immediately monitored at 510 nm with excitations at both 460 and 403 nm recorded simultaneously for 300 s using fluorescence spectrophotometer (Hitachi, Model F-7100, Japan). At 300 s, an aqueous solution of Triton X-100 (20 μL, 20% v/v) was immediately added to induce the maximum change in fluorescence emission. The final transport trace was obtained as a ratiometric value of $I_{460}/I_{403}$ and normalized based on the ratiometric value of $I_{460}/I_{403}$ after addition of triton using the Eq. (1).

$$I_f = (I_t - I_0)/(I_1 - I_0) \tag{1}$$

where $I_f$ = Fractional emission intensity, $I_t$ = Fluorescence intensity at time t, $I_1$ = Fluorescence intensity after addition of Triton X-100 and $I_0$ = Initial fluorescence intensity.

### CF dye leakage assay
Preparation of CF-containing LUVs follows a similar protocol in the HPTS assay. In a typical experiment, the CF-containing LUV suspension (30 μL, 13 mM in 10 mM HEPES buffer containing 100 mM NaCl at pH = 7.5) was diluted to the same HEPES buffer solution (1.75 mL) to create a concentration gradient for CF dye efflux study. A solution of channels or Melittin in DMSO at different concentrations was injected into the suspension under gentle stirring. Upon the addition of channels or melittin, the emission of CF was immediately monitored at 517 nm with excitations at 492 nm for 300 s using fluorescence spectrophotometer (Hitachi, Model F-7100, Japan). At $t$ = 300 s, an aqueous solution of Triton X-100 (20 μL, 20% v/v) was immediately added to achieve the maximum efflux of dye. The final transport trace was obtained by normalizing the fluorescence intensity using Eq. (2).

$$F = (F_t - F_0)/(F_1 - F_0) \tag{2}$$

where, $F$ = Fractional emission intensity, $F_t$ = Fluorescence intensity at time $t$, $F_1$ = Fluorescence intensity after addition of Triton X-100 and $F_0$ = Initial fluorescence intensity.

The fractional ion transport activity R was calculated for each curve using the normalized fluorescence intensity before the addition of Triton X-100, with the blank as 0 and that of triton set 1. Fitting the fractional transmembrane activity R vs channel concentration using the Hill Eq. (3) gave the Hill coefficient n and $EC_{50}$ values.

$$R = 1/(1 + (EC_{50}/[channel])^n) \tag{3}$$

### Single channel conductance measurement using planar lipid bilayer system
The chloroform solution containing a mixture of 1,2-diphytanoyl-sn-glycero-3-phosphocholine (diPhyPC, 10 mg/ml, 40 μL) and cholesterol

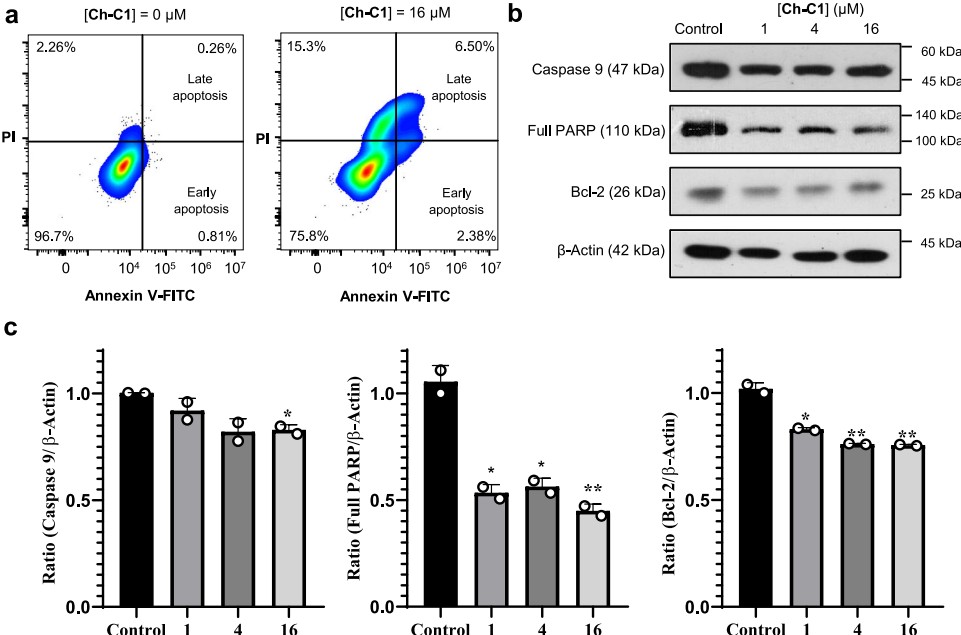

**Fig. 5 | Nanopores induce cancer cell apoptosis. a** Evaluation of HepG2 apoptosis by flow cytometry, with **Ch-C1** at 0 μM and 16 μM for 24 h and cells stained using green Annexin V-FITC conjugate and red PI dye. Annexin V = Intracellular protein of the annexin family that recognizes phosphatidylserines; FITC = Fluorescein isothiocyanate. **b** Immunoblot assay for Caspase 9, PARP and Bcl-2 in HepG2 cells treated with up to 16 μM of **Ch-C1** for 24 h. Results were analyzed via ImageJ 1.8.1 and reported as histograms by graphpad prism 8.01 in **c**. Two-tailed Student's *t*-test was used for statistical significance and data are presented as means ± SEM (*n* = 2 biologically independent samples). *P* values (Caspase 9) = 0.1827, 0.0524 and 0.0105, respectively. *P* values (full PARP) = 0.0132, 0.0156 and 0.0093, respectively. *P* values (Bcl-2) = 0.0117, 0.0060 and 0.0058, respectively. Symbols * and ** stand for significant differences between the control group and other groups, with *P* < 0.05 and 0.01, respectively. Source data are provided as a Source Data file.

(10 mg/ml, 9.1 μL) was evaporated using nitrogen gas to form a thin film and re-dissolved in n-decane (16 uL). 0.5 μL of this n-decane solution was injected into the aperture of the Delrin® cup (Warner Instruments, Hamden, CT) with the n-decane removed using nitrogen gas. In a typical experiment, both the chamber (*cis* side) and Delrin cup (*trans* side) were filled with an aqueous KCl solution (1.0 M, 1.0 mL). Ag-AgCl electrodes were inserted into the two solutions with the *cis* chamber grounded. Planar lipid bilayer was formed by painting 0.3 μL of the lipid-containing n-decane solution around the n-decane-pretreated aperture. Samples in MeOH (1.0 μL) were added to the *cis* compartment to reach a final concentration of around $10^{-6}$ M and the solution was stirred for a few min until a single current trace appeared. These single channel currents were then measured using a Warner BC-535D bilayer clamp amplifier, collected by PatchMaster (HEKA) with a sample interval at 5 kHz and filtered with an 8-pole Bessel filter at 1 kHz (HEKA). The data were analysed by FitMaster (HEKA) with a digital filter at 100 Hz. Plotting current traces vs voltages yielded ion conductance (γ). The diameter of the channel is calculated from the ion conductance using Hille Eq. (4):

$$1/g = (l + \pi d/4) \times (4\rho/(\pi d^2)) \tag{4}$$

where *g* = corrected ion conductance (obtained by multiplying measured conductance with Sansom's correction factor), *l* = length of channel (40 Å) and *ρ* = resistivity of 1 M KCl solution (0.0921 Ω•m).

### In vitro anticancer study via MTT assay

The cytotoxicity assay of channels relied mainly on ((3-(4,5-dimethylthiazol-2-yl)-2,5-diphenyl tetrazolium) MTT assay in which MTT was used as the detection reagent for cell viability. HepG2, U87-MG, THLE-2 and HK-2 were used as test cells. Cells were cultured in Dulbecco's Modified Eagle Medium (DMEM, Gibco), containing 10% of fetal bovine serum (FBS, Gibco) and 10,000 U/mL of Penicillin-Streptomycin (Gibco) at 37 °C in a humidified atmosphere containing 5% $CO_2$.

Active cells were seeded onto 96-well plates with a cell density of $1 \times 10^4$ cells/well in 100 μL DMEM medium. Channel molecules at a series of diluted concentrations in DMSO (obtained from Yeasen) were further diluted with the serum-free medium to give final concentration of channel molecules ranging from 0.0625 μM to 512 μM. The cells treated with 0.5% DMSO served as a negative control. After the cell density reached 50–60%, the medium of 96-well plates was replaced with serum-free medium containing channel reagents of various concentrations, and each concentration was repeated four times. After 48 h incubation, the medium containing the channel molecules were replaced with MTT solution (free DMEM as solvent, 500 μg/mL). Then after 4 h incubation, the medium was removed, and 150 μL DMSO were added, followed by uniformly shaking the plates to completely dissolve the purple formazan crystals. The UV-absorption values at 490 nm were obtained by absorbance microplate reader (CMAX PLUS, Molecular Devices, USA). Cell viabilities with the addition of channel molecules at various concentrations were calculated following Eq. (5). Then cell viabilities versus logarithm of channel concentrations were plotted and the IC50 values were calculated using a nonlinear regression curve fit with Graphpad Prism 8.0.1.

$$\%\text{Cell Viability} = (OD_{490}(\text{channel})/OD_{490}(5\%DMSO)) \times 100\% \tag{5}$$

### Cell membrane integrity assessment

HepG2 cells were seeded onto a 48-well plate with pre-placed circular cell plates at cell density of $2 \times 10^4$ cells/well. After incubation at 37 °C in a humidified atmosphere containing 5% $CO_2$ for 12 h, the supernatant was replaced with the medium containing **Ch-C1** at 0, 1, 4 and 16 μM, and cell were incubated for another 36 h. The propidium iodide (PI) staining solution (obtained from Yeasen) was diluted 20 times with dye diluent and warmed up to 37 °C in a water bath. The supernatant of the 48-well plate was then replaced with the warm PI staining solution. After incubation for 15 min, the cells were washed twice with phosphate buffered

saline (PBS, Solarbio) and fixed at 4 °C with a 4% paraformaldehyde solution for 10 min. The tablets were sealed with 4',6-diamidino-2-phenylindole (DAPI)-containing cell sealing tablets (obtained from Yeasen) and dried naturally in the dark. The fluorescence images were then photographed using laser confocal microscope (LSM5 EXCITER, Zeiss).

## Flow cytometry assay

HepG2 cells were cultured in 6-well plates containing DMEM supplemented with 10% fetal bovine serum at 37 °C in a humidified atmosphere containing 5% $CO_2$ for 12 h. After incubating the cells with **Ch-C1** at various concentrations for 24 h, the cells were detached from the plate using 0.25% trypsin (0.5 mL) and resuspended in medium (4 mL) prior to centrifugation (112 × g for 5 min). The obtained cell pellets were washed twice using PBS (2 mL, Solarbio) and stained with Annexin V-FITC (obtained from Yeasen) and propidium iodide (PI) in the dark for 15 min. The percentages of apoptotic cells were determined by flow cytometry using PB450-A (CytoFLEX, Beckman Coulter) for 1 h. Data was analyzed using FlowJo 10.6.2. Example of gating strategy for apoptotic cell analysis is illustrated in Supplementary Fig. 42.

## Immunoblot analysis

HepG2 cells were evenly seeded in 6-well plates with cell density of $6 \times 10^5$ cells/well and incubated in the medium containing FBS at 37 °C in a humidified atmosphere containing 5% $CO_2$ for 16 h, followed by the replacement of medium containing **Ch-C1** at 0, 1, 4 and 16 μM without FBS. Meanwhile, control cell groups were prepared via the same protocol using 0.5% DMSO in the absence of **Ch-C1**. After incubation with **Ch-C1** for 24 h, the cells were washed with PBS once, and lysed in a lysis buffer containing 250 mM NaCl, 20 mM Tris-HCl at pH 7.4, 1.0 mM ethylene diamine tetraacetic acid (EDTA, Solarbio) and 1% Triton X-100 (Adamas) at 4 °C for 30 min. Protein gel electrophoresis was performed using sodium dodecyl sulfate polyacrylamide gel electrophoresis (SDS-PAGE) method. The separated proteins were then transferred to polyvinylidene fluoride (PVDF, Millipore) membrane. After blocking with 5% skimmed milk, blots were incubated in primary antibody solution overnight at 4 °C. Antibodies used are sc-8007 (1:1000, Santa) for PARP, 9502 T for Caspase 9 (1:1000, Cell signalling technology), 12324 for BCL2 (1:1000, Abcam) and AC026 for β-Actin (1:5000, Abclonal). After washing with TBST, incubation continued with a peroxidase-conjugated secondary antibody solution (1:5000, Abcam) at 25 °C for 2 h, and the blots were then washed with 1xTBST solution, developed with Pierce™ ECL Western Blotting Substrate (Thermo Scientific) and visualized using Chemiluminescence Imager (ChemiDoc MP, Biorad). Density of each band was analysed with ImageJ 1.8.0 software. Uncropped scans of blots are supplied in the Source Data file.

## Reporting summary

Further information on research design is available in the Nature Research Reporting Summary linked to this article.

# Data availability

The authors declare that the data supporting the findings of this study are available within the paper and its supplementary information files. The datasets that support the finding of this study are available in figshare repository with the identifier(s) https://doi.org/10.6084/m9.figshare.20544585.v1. Source data are provided with this paper.

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

## Acknowledgements

This work was supported by the National Natural Science Foundation of China (22001221 to C.R., 81971724 to Y.-L.W and U1903119 to C.W.), Shenzhen Science and Innovation Committee (JCYJ20210324123411030 to C.R. and 2021Szuvp067 to C.R.), the Fundamental Research Funds for the Central Universities (20720210101 to C.R.) and Fuzhou University.

## Author contributions

J.S. performed the synthesis and ion transport study of channels. Y.G. performed the single channel current measurement. L.K., C.W. and Y.-L. W. performed in vitro anticancer study and hemolytic study. Q.Z. performed flow cytometry study and membrane integrity study. Y.C. and Z.W. performed immunoblot study. Y.M. performed the computational work. Y.L. performed the purification of channels. C.R. and H.Z. conceived and supervised the project and wrote the manuscript. All authors participated in discussion and editing of the manuscript.

## Competing interests

The authors declare no competing interests.
