## [Peer Review File · Nature Communications]

Cholesterol-stabilized membrane-active nanopores with anticancer activitiesReviewers' comments:

Reviewer #1 (Remarks to the Author):

This manuscript describes the cholesterol-dependent pore formation of synthetic molecules that has two cholic acids at both ends. Fluorescence assay using liposomes confirmed that the pore-forming activity of the molecules increases as the cholesterol content in the lipid bilayer ranging from 0 - 50 mol%. The pore size was evaluated by fluorescence leakage assay and channel current measurements using planar lipid bilayers, and it was found that the leaking and current amplitudes were changed with changing the length of the side chains. It was also confirmed that the designed molecule has high toxicity to cancer cells.

Although the behavior of the pore-formation of this synthetic molecule may be interesting in terms of supramolecular chemistry, this report has many concerns which should be addressed, as I listed below. Therefore, I recommend that the author should make the revised manuscript for the resubmission.

1. This is crucial. There is no experimental evidence to form nanopore structure by assembling your molecules with cholesterol as you presented in Fig.1c. I understood that this molecule can transport both CF dye and ions based on the results from the liposome leaking assay and the channel current measurements. However, these experimental results do not ensure the assembling of pore-formation with cholesterol. You should perform additional experiments to confirm the pore-formation; for example, the spectroscopic experiments reported by another group (Nat. Commun. 2020, 10.1038/s41467-020-16770-z) may be useful.
 2. Regarding the novelty, the structure of the presented molecules is similar to the author's previous report in Angew 2019. This molecule also has two cholic acids and a flexible side chain with a crown ether which swings to bring alkali metal ions, as mimicking an ion pump. This is quite interesting report. Besides, the reported channel activity also has cholesterol-dependency. Considering the author's previous research, this report may have not so high novelty and originality.
 3. The introduction is too short to understand the previous and the current research situation on the synthetic pore or channel. How is the previous report on the cholesterol-dependent channels? How is the strategy for constructing the pore??
 4. In the anticancer activity, the mechanism of the activity is unclear. There is no experimental evidence of the killing cell mechanism (membrane disruption or the other mechanism such as the inhibition of some enzymes). Also, you should take a control experiment of the cytotoxicity using a red blood cell or a normal cell line.
 5. There is a lot of errors and a lack of the experimental details below.
- 5-1. The caption of Fig. 2 is uncompleted.

5-2. A more detailed description of the MD simulation is needed, such as the initial model, the calculation time, and the temperature.

5-3. There are no error bars in Fig. 2b, 2c, 3a, 3b, and S13. Also, how did you determine the specific current value in Fig. 2a and 2b because the open current level is not constant in your current trace (Fig. 2a)?

5-4. The eq. (3) seems to be not correct, and the resistivity of 1 M KCl solution maybe not correct.

5-5. There are many mistakes in the References. Please check them out.

5-6. There are many other lacking and mistakes. Please check carefully over the entire manuscript including SI by several authors.

Reviewer #2 (Remarks to the Author):

Cholesterol-assisted poration of membranes is widely utilized by bacterial cells to specifically target eukaryotic cells, whilst leaving their own membranes free from toxicity. In this manuscript, the authors present an artificial cholesterol-dependent nanopore that shows size dependent small molecule flux and displays anticancer activity.

Overall, the manuscript is written coherently and there is novelty in the nanopore constructed here. However, there are a few points that should be addressed before this paper is accepted for publication.

Firstly, I am not sure mechano-nanopore is the correct terminology for these pores – no experiments have been conducted to determine their activation in response to mechanical stimuli (e.g. bacterial MS channels that are activated in response to changes in osmotic pressure are mechanosensitive channels).

Secondly, detailed illustrations have been produced showing the alternating pore-cholesterol configuration, but no experiments have been carried out to show how many cholesterols form to make the pore. There should be some data on the structure either via experiments or simulations.

Thirdly, there are some problems with the single-channel current recording data and analysis that need to be addressed. It is stated in the text that the pore likely takes a “barrel-stave” model. This is a highly regular arrangement of subunits that results in clean regular step-like current changes, for example in the case of Alamethicin¹. However, the example traces in Figure 3a (with exception of -20 and -60 mV) show incredibly noisy insertions that do not fit the barrel-stave model – and I would further question whether the traces at 100 mV, 80 mV, -40 mV and -100 mV are actual nanopore insertions at all – they look like non-pore related membrane disruptions. More example traces are needed that fit the barrel-

stave model at a wider range of voltages. Otherwise, the authors have to modify their claim that this is a barrel-stave pore.

In addition to this, the IV curves in Figure 3a and b have no error bars. I would expect large errors, especially at the high voltages (>50 mV) considering the wide current variation of example traces in Figure 3a and histograms shown in the SI. The IV curves also have no associated n numbers in the figure legend or text, these must be included to show how many individual pore insertions the IV curves are representative of. The conductance error is also surprisingly low, considering the aforementioned current variation seen in Figure 3a and SI figures. Again, n numbers need to be included here.

Lastly, there is limited information on the nanopores anti-cancer activity. A graphical figure showing concentration dependent cell death in comparison to control cells and known anticancer drugs would greatly improve this section. Further experiments showing cell-binding with fluorescent tags would also improve the validity of this section. From the data it is not know whether the nanopores are disrupting the cell membrane, or are being internalized and disrupting intracellular membranes.

(1) Haris, P. I.; Molle, G.; Duclohier, H. Conformational Changes in Alamethicin Associated with Substitution of Its α -Methylalanines with Leucines: A FTIR Spectroscopic Analysis and Correlation with Channel Kinetics. *Biophys. J.* 2004, 86 (1 I), 248–253. [https://doi.org/10.1016/S0006-3495\(04\)74100-4](https://doi.org/10.1016/S0006-3495(04)74100-4).

Comments by Reviewer 1 and Our Responses

Overall comments: This manuscript describes the cholesterol-dependent pore formation of synthetic molecules that has two cholic acids at both ends. Fluorescence assay using liposomes confirmed that the pore-forming activity of the molecules increases as the cholesterol content in the lipid bilayer ranging from 0 - 50 mol%. The pore size was evaluated by fluorescence leakage assay and channel current measurements using planar lipid bilayers, and it was found that the leaking and current amplitudes were changed with changing the length of the side chains. It was also confirmed that the designed molecule has high toxicity to cancer cells.

Although the behavior of the pore-formation of this synthetic molecule may be interesting in terms of supramolecular chemistry, this report has many concerns which should be addressed, as I listed below. Therefore, I recommend that the author should make the revised manuscript for the resubmission.

Comment 1: This is crucial. There is no experimental evidence to form nanopore structure by assembling your molecules with cholesterol as you presented in Fig.1c. I understood that this molecule can transport both CF dye and ions based on the results from the liposome leaking assay and the channel current measurements. However, these experimental results do not ensure the assembling of pore-formation with cholesterol. You should perform additional experiments to confirm the pore-formation; for example, the spectroscopic experiments reported by another group (Nat. Commun. 2020, 10.1038/s41467-020-16770-z) may be useful.

Our response: As suggested, we have carried out three sets of additional experiments, i.e., ¹H NMR titration, UV-vis titration and mass spectrometry on channel molecules in the absence and presence of cholesterol molecules. Especially from HR-MS experiments, we believe two channel molecules may self-dimerize to form unit A, which then interacts with a tail-to-tail assembled cholesterol unit B made up of two cholesterol molecules to generate a (AB)_n type barrel-stave pore (Figure 3c). Molecular dynamics simulation shows that pores generated with n = 5 enclose pore sizes of 1.37 nm and 0.80 nm comparable to experimentally determined 1.48 nm and 0.97 nm for **Ch-C1** and **C1-C4**. Certainly, other molecular ratios between channel and cholesterol molecules can't be ruled out completely. One such alternative example, i.e., (**Ch-C1•2Ch**)₁₀ of 1.40 nm in diameter was presented in Figure 3c.

The following *italicized* Sections have been added into the main text (Pages 4 and 5), with Figure S17 (¹H NMR titration), Figures S18-S19 (UV-vis spectra) and Figures S20-S24 (mass spectrometry) added into SI.

Pore-formers do associate with cholesterol molecules

*The binding between the channel molecules and **Ch** was initially supported by ¹H NMR titration experiments involving titrating 0 – 100 equivalents of **Ch** into a THF-d₈ solution containing **Ch-C2** at 5 mM (Figure S17). Upon addition of up to 100 equivalents of **Ch**, the signals corresponding to the two amide protons of **Ch-C2** show obvious downfield shifts of up to 0.08 and 0.16 ppm, respectively, indicating that not only **Ch-C2** can bind to **Ch** molecules but also such binding likely increases the self-association extent of **Ch-C2** molecules. Using UV-vis spectroscopy, we observed (1) red shifts of 2.5 nm and 21 nm for the maximum and minor absorption peaks when increasing the concentration of **Ch-C2** from 5 μM to 80 μM in THF at 20 °C (Figure S18) and (2) a red shift of 2.5 nm and a 24% decrease in intensity for the maximum absorption peak (Figure S19). These two pieces of data are evidently*

suggestive of associations among **Ch-C2** molecules and between **Ch-C2** and **Ch** molecules. Consistent with the ¹H NMR- and UV-vis-based binding data, analysis of the high-resolution mass spectra (HRMS) of the THF solution containing **Ch-C2** (10 μM) and **Ch** (1 mM) reveals dimeric (**Ch-C2**)₂•Na⁺ and (**Ch-C2**)₂•H⁺ as well as 4**Ch-C2**•2**Ch**•2H⁺ as the major peaks (Figures S21-S22), followed by much weaker signals, corresponding to the singly charged (**Ch-C2**)₂•H⁺ that associates with one or two **Ch** molecules (Figures S23-S24). These HRMS data suggest self-dimerization involving two **Ch-C2** molecules to be far much stronger than the mutual-association between **Ch-C2** and **Ch** molecules.

Computational models of the pores formed by Ch-C1 and Ch-C4

In light of (1) the side chain-dependent activity trend deduced from the CF leakage assay, (2) the quantified pore sizes of 1.48 and 0.97 nm for **Ch-C1** and **Ch-C4**, respectively, and (3) the HRMS data that confirm the existence of a dimeric (**Ch-C1**)₂ fragment, we constructed a few barrel-stave ensembles [(**Ch-Cn**)₂•2**Ch**]_m, with side chains pointing toward the pore interior (Figures 3c and S25a). This was followed by molecular dynamics simulation (MD) to yield pore sizes of 1.37 and 0.80 nm for [(**Ch-C1**)₂•2**Ch**]₅ and [(**Ch-C4**)₂•2**Ch**]₅, respectively. Moreover, ensembles [(**Ch-Cn**)₂•2**Ch**]_m with m ≠ 5 generate a pore size that is too small or too large (Figure S25a), and those with half of side chains pointing outward are energetically less stable (Figure S25b). We thus believe pentameric pores containing inward-pointing side chains might be one preferred association mode at least for **Ch-C1** and **Ch-C4**, enabling channel and **Ch** molecules to act synergistically to produce membrane-active wide pore ensembles. Certainly, there might exist other possible structural models that may also enclose a pore size comparable to the experimentally determined one. Figure 3d illustrates one such ensemble (**Ch-C1**)₂•2**Ch**]₁₀, having a pore size of 1.40 nm that closely matches the pore size of 1.48 nm for **Ch-C1**.

Comment 2: Regarding the novelty, the structure of the presented molecules is similar to the author's previous report in Angew 2019. This molecule also has two cholic acids and a flexible side chain with a crown ether which swings to bring alkali metal ions, as mimicking an ion pump. This is quite interesting report. Besides, the reported channel activity also has cholesterol-dependency. Considering the author's previous research, this report may have not so high novelty and originality.

Our response: On the one hand, regarding the novelty issue, the following **Figure A** illustrates our earlier work in Angew 2019, which is termed molecular swing that applies a crown ether to swing ions across the membrane. And it transports ONLY cations. **Figure B** presents our current design that applies only the linear scaffold component, having such as a methyl group to replace the crown ether-containing swinging component in **Figure A**. This series of molecules then form a barrel-stave pore and transport cations, anions and small molecules, uniquely with transport activities up-regulated by cholesterol. Therefore, in terms of structure, types of molecular species transported and difference in transport mechanism (e.g., swinging vs channel mechanisms), the novelty of our work is obvious.

On the other hand, for the cholesterol-dependent ion transport activity, our earlier work in Angew 2019 indeed is cholesterol-dependent, but it becomes less and less active in the presence of increasing amounts of cholesterol. In fact, this cholesterol-attenuated activity is invariably observed in all artificial channels/carriers reported over the past four decades. Therefore, our current work truly presents the first unprecedented example with ion transport activity becoming increasingly active in the presence of increasing amount of cholesterol. This has been discussed in our introduction.

Comment 3: The introduction is too short to understand the previous and the current research situation on the synthetic pore or channel. How is the previous report on the cholesterol-dependent channels? How is the strategy for constructing the pore??

Our response: As suggested, we have added the following *italicized* paragraph into the Section of “**Molecular design of pore-forming Ch-Pns**” (Pages 1 and 3), citing different pore-forming strategies by Moore, Matile, Kobuke, Regen, Gong, Muraoka and Kinbara, Zeng, Talukdar, Davis, Dash and Wanunu. This was not added into the introduction because we believe a short and concise introduction is better and not too distractive as we want emphasize the unique feature of our channels that exhibit a cholesterol-enhanced transport activity. Further, to our best knowledge, all artificial membrane transports exhibit cholesterol-attenuated properties, and this statement has been made in the introduction.

Results and Discussion

Molecular design of pore-forming Ch-Pns

Attracting wide interests from biomimetic chemists, artificial transmembrane pores can be readily constructed using backbone-rigidified foldamers^{48,67,79-81}. We however envisioned that this type of pores with high structural rigidity unlikely will exhibit cholesterol-enhanced ion transport activities. Instead, those pore-enclosing conformations, which are generated from either a single molecular backbone without a defined conformation^{9,20,28,40,65} or multiple components assembled via non-covalent forces, are deemed to be more sensitive toward environmental stimuli. In this regard, there exist a wide range of strategies in the literature. Inspiring ones include helical folding via solvophobic forces by Moore⁹, “rigid rod” β -barrels by Matile¹³⁻¹⁵, tail-to-tail assembly of half channel molecules¹⁶ and pore-forming steroid-modified biscarbamate¹⁷ by Kobuke, dendritic¹⁹ or linear-shaped steroid derivatives¹⁹ by Regen, columnar stacking involving shape-persistent macrocycles by Gong^{23,27,49}, multiblock amphiphiles having alternately arranged hydrophilic and hydrophobic units by Muraoka and Kinbara^{28,40,65}, structurally simple pore-forming mono- and tri-peptides^{71,72} and trimesic amides⁷³ by Zeng, as well as H-bond-assisted formation of pore-

containing rosettes derived from mannitol³⁶ or diol-containing 1,3-diethynylbenzene⁵² by Talukdar, folate by Matile¹⁸, G-quartets by Davis²² and Dash^{44,66} and a fused guanine–cytosine base by Wanunu⁶⁹.

Comment 4: In the anticancer activity, the mechanism of the activity is unclear. There is no experimental evidence of the killing cell mechanism (membrane disruption or the other mechanism such as the inhibition of some enzymes). Also, you should take a control experiment of the cytotoxicity using a red blood cell or a normal cell line.

Our response: We have carried out additional experiments to address all the concerns raised by the Reviewer. The following *italicized* discussions have been added into Pages 6 and 7 of the main text.

*These good anticancer activities of **Ch-C1** promoted us to further evaluate its selectivity and safety towards normal cells by performing the cell viability assay on human normal liver cells (LO2 cell line) and human embryonic kidney cells (HEK 293T cell line). Desirably, the determined IC₅₀ values of **Ch-C1** against LO2 and 293T cells were 70.3 and >500 μM, respectively (Figure S28). Defined as the ratio of IC₅₀ values between normal and cancer cells, the selectivity indexes (SI) determined for **Ch-C1** molecules are 18.5 and > 130 against LO2 and 293T cells, respectively. These high SI values are more than one order of magnitude higher than those of doxorubicin and paclitaxel⁸⁹⁻⁹⁰, indicating comparably lower cytotoxicities of **Ch-C1** to normal cells than doxorubicin and paclitaxel. We then employed mice blood cells to test the *in vitro* hemolytic activity of **Ch-C1**. To our delight, **Ch-C1** also shows a low hemolytic activity. The concentration causing 50% hemolysis of red blood cells (HC₅₀) is 127.3 μM (Figure S29), corresponding to a high SI value of 38.5. This high specificity may be caused by the high level expression of cholic acid receptors on hepatocarcinoma cells that leads to higher uptake of cholic acid-containing channel molecules⁹¹⁻⁹².*

*Two types of dye molecules (PI and DAPI) were applied to assess the membrane integrity in the presence of **Ch-C1** molecules. While DAPI molecules can enter the cells at high concentrations and become blue upon binding to the AT regions of dsDNA, membrane-impermeable PI enters cells having compromised membranes and binds tightly to the intracellular nucleic acids to emit red fluorescence when excited at 535 nm. After the incubation of HepG2 cells with various concentrations of **Ch-C1** (0, 1, 4 and 16 μM) for 36 h, the cells were fixed, stained with PI and DAPI, and analyzed under a laser confocal microscope. As can be seen from Figure 4, the DAPI-stained cells reveal intact cell membrane, confirming that pore-forming **Ch-C1** does not disrupt the cell membranes to a noticeable extent at concentrations of up to 16 μM. And the PI-stained cells do suggest that **Ch-C1** make the membrane leaky and more permeable via forming wide pores.*

Figure 4. Cell imaging of HepG2 cells treated with **Ch-C1** at concentrations of 1 μM, 4 μM and 16 μM for 36 h, followed by staining with blue DAPI and red PI dyes. Blue and red images were merged by Image J. DAPI = 4',6-diamidino-2-phenylindole; PI = propidium iodide.

Figure 5. (a) Evaluation of HepG2 apoptosis by flow cytometry, with **Ch-C1** at 0 μM and 16 μM for 24 h and cells stained using green Annexin V-FITC conjugate and red PI dye. Annexin V = Intracellular protein of the annexin family that recognizes phosphatidylserines; FITC = Fluorescein isothiocyanate. (b) Immunoblot assay for Caspase 9, PARP and Bcl-2 in HepG2 cells treated with up to 16 μM of **Ch-C1** for 24 h. Results were analyzed via Image J and reported as histograms by graphpad prism 8.01 in (c). Symbols * and ** stand for significant differences between the control group and other groups, with $P < 0.05$ and 0.01, respectively.

It has been shown that disruption of cellular homeostasis caused by leaky cell membrane leads to cell apoptosis via the caspase signaling pathway⁹³⁻⁹⁵. Applying dead cell apoptosis kit, we treated the HepG2 cells with **Ch-C1** at 16 μM for 24 h, stained the cells using both green Annexin V-FITC conjugate and red PI dyes and sorted the cells by flow cytometry. Considering that cells stainable by Annexin V-FITC conjugate correspond to early or later apoptotic cells (Figure 5a), the fact that the percentage of apoptotic cells substantially increases from 1.07% to 8.88% with increased concentrations of **Ch-C1** from 0 to 16 μM establishes the capability of **Ch-C1** to induce cell apoptosis. Some characteristic proteins involved in apoptosis include (1) apoptosis initiator protein (caspase-9) that initiates cell killing but is cleaved during early apoptosis^{96,97}, (2) poly(ADP-ribose) polymerase (PARP) cleaved by the activated caspase 9 to facilitate apoptosis by preventing DNA repair and (3) anti-apoptotic protein (Bcl-2) that undergoes apoptosis-induced inhibition. Therefore, following an apoptotic stimulus, decreased presences of caspase 9, PARP and Bcl-2, together with the increased presence of cleaved PARP, are expected. And these are indeed what we observed when we treated HepG2 cells with **Ch-C1** at 16 μM for 6 h (Figure 5b,c), prompting us to conclude that **Ch-C1**-enhanced membrane permeability can induce HepG2 cell apoptosis via the caspase 9 pathway.

Comment 5: There is a lot of errors and a lack of the experimental details below.

5-1. The caption of Fig. 2 is uncompleted.

Our response: Figure 2 caption has been made complete.

5-2. A more detailed description of the MD simulation is needed, such as the initial model, the calculation time, and the temperature.

Our response: The structural models were built using Gaussian 09, and then optimized using the generalized amber force field (GAFF). The simulation continues for 200 ns at 300 K. These points have been added into Page S36 as shown below.

5-3. There are no error bars in Fig. 2b, 2c, 3a, 3b, and S13. Also, how did you determine the specific current value in Fig. 2a and 2b because the open current level is not constant in your current trace (Fig. 2a)?

Our response: As suggested, we have added error bars for Figs. 2b, 3a, 3b, and S16 (but not Fig 2c as adding error bars of less than 3% makes the figure too messy, but we do mention the relative errors in the figure caption).

As to “how did you determine the specific current value in Fig. 3a and 3b” (Fig 2a and 2b should be the type errors by the Reviewer), we have explained it in Page 4 of the main text. That is, we plotted the histogram of currents based on with a mean current was obtained. Take single channel current traces recorded at 80 mV for **Ch-C1** as the example, we first obtained its histogram by plotting the counts vs current magnitudes in pA. The current (16.2 pA) of the most dominant peak having the most counts was then taken as the mean current value. The relative error was obtained by dividing the width between the two half-height points by 2 (e.g., 10.0 pA / 2 = 5.0 pA in this case).

For other histograms, see Figures S12, S13 and S16.

5-4. The eq. (3) seems to be not correct, and the resistivity of 1 M KCl solution maybe not correct.

Our response: Based on the value reported in *J. Res. Natl. Inst Stand. Technol.* **1994**, 99, 241-246, we have changed the resistivity value for 1 M KCl solution from 0.0947 $\Omega\cdot\text{m}$ to 0.0921 $\Omega\cdot\text{m}$. The calculated pore sizes for **Ch-C1** and **Ch-C4** were changed to 1.48 nm and 0.97 nm, respectively.

5-5. There are many mistakes in the References. Please check them out.

Our response: We have corrected all mistakes in the Ref Section.

5-6. There are many other lacking and mistakes. Please check carefully over the entire manuscript including SI by several authors.

Our response: We have tried our best to make all necessary changes with some shown below:

“hours” changed to “h”

“were” changed to “was”

“ $\delta=$ ” changed to “ $\delta =$ ”

“4°C” changed to “4 °C”

“vacuo” changed to “*vacuo*”

“0.5 mmol” changed to “0.50 mmol”

“1.0 mmol” changed to “1.00 mmol”

“at a concentration of” changed to “at concentration of”

“at pH = 8.0” changed to “at pH 8.0”

font size change in page S14

“100 mL x 3” changed to “3 x 100 mL”

“flash column” changed to “flash column chromatography”

double space replaced by single space

“**2014,**” changed to “**2014,**” in the ref section

“Fluorescence intensity changed” changed to “Fluorescence intensity changes”

Comments by Reviewer 2 and Our Responses

Overall comments: Cholesterol-assisted poration of membranes is widely utilized by bacterial cells to specifically target eukaryotic cells, whilst leaving their own membranes free from toxicity. In this manuscript, the authors present an artificial cholesterol-dependent nanopore that shows size dependent small molecule flux and displays anticancer activity.

Overall, the manuscript is written coherently and there is novelty in the nanopore constructed here. However, there are a few points that should be addressed before this paper is accepted for publication.

Comment 1: Firstly, I am not sure mechano-nanopore is the correct terminology for these pores – no experiments have been conducted to determine their activation in response to mechanical stimuli (e.g. bacterial MS channels that are activated in response to changes in osmotic pressure are mechanosensitive channels).

Our responses: We agreed with the Reviewer and have removed the term “mechano” throughout the manuscript. The title has also be revised accordingly as shown below:

Original title: Cholesterol-Stabilized Membrane-Active Mechano-Nanopores with Potent Anticancer Activities

Revised title: Cholesterol-Stabilized Membrane-Active Nanopores with Potent and Specific Anticancer Activities

Comment 2: Secondly, detailed illustrations have been produced showing the alternating pore-cholesterol configuration, but no experiments have been carried out to show how many cholesterol form to make the pore. There should be some data on the structure either via experiments or simulations.

Our response: As suggested, we have carried out three sets of additional experiments, i.e., ¹H NMR titration, UV-vis titration and mass spectrometry on channel molecules in the absence and presence of cholesterol molecules. Especially from HR-MS experiments, we believe two channel molecules may self-dimerize to form unit A, which then interacts with a tail-to-tail assembled cholesterol unit B made up of two cholesterol molecules to generate a (AB)_n type barrel-stave pore (Figure 3c). Molecular dynamics simulation shows that pores generated with n = 5 enclose pore sizes of 1.37 nm and 0.80 nm comparable to experimentally determined 1.48 nm and 0.97 nm for **Ch-C1** and **C1-C4**. Certainly, other molecular ratios between channel and cholesterol molecules can't be ruled out completely. One such alternative example, i.e., (**Ch-C1•2Ch**)₁₀ of 1.40 nm in diameter, was presented in Figure 3c.

Comment 3: Thirdly, there are some problems with the single-channel current recording data and analysis that need to be addressed. It is stated in the text that the pore likely takes a “barrel-stave” model. This is a highly regular arrangement of subunits that results in clean regular step-like current changes, for example in the case of Alamethicin¹. However, the example traces in Figure 3a (with exception of -20 and -60 mV) show incredibly noisy insertions that do not fit the barrel-stave model – and I would further question whether the traces at 100 mV, 80 mV, -40 mV and -100 mV are actual nanopore insertions at all – they look like non-pore related membrane disruptions. More example traces are needed that fit the barrel-stave model at a wider range of voltages. Otherwise, the authors have to modify their claim that this is a barrel-stave pore.

Our response: We believe that, assisted by the cholesterol molecules, barrel-stave pores are indeed formed. This can be supported by (1) regular step-like current changes at -20 mV and -80 mV in **Figure A**, (2) a nearly linear fitting of I-V curve in **Figure B** and (3) examples shown in **Figure C** and **D**. Nevertheless, the pore formation through non-covalent forces highly likely is a dynamic process, generating a mixture of nanopores, having different diameters or different channel over cholesterol molar ratios. And even the same nanopore may not have a fixed pore size given that (1) non-covalent associations between channel and cholesterol molecules are not that tight as inferred from HR-MS spectra, (2) lipid molecules are in constant movement that exerts varied interacting forces onto the nanopore and its components, altering the pore size from time to time, and (3) applied voltage may also influence the self-assembling process. This can explain why we observe irregular current traces at 100 mV and 80 mV, and more regular current traces at 60 mV, -40 mV, -80 mV and -100 mV in **Figure A**.

In **Figure A**, the red numbers refer to the mean current values obtained from their histograms presented in Figures S12 and S13, and subsequently were used to plot I-V curve shown in **Figure B** for obtaining the conductance value.

Figure C and **D** illustrate two examples of single channel current traces obtained using carbon nanotubes and the metal complexes as the transmembrane ion channels, respectively.

Regarding “More example traces are needed that fit the barrel-stave model at a wider range of voltages”, we indeed have attempted to obtain current traces at such as 140 mV, but the cholesterol-containing lipid bilayer for single channel current measurement seems to become unstable and ruptured (despite of the fact that we do know that cholesterol enhances the stability of LUVs and cell membrane).

Comment 4: In addition to this, the IV curves in Figure 3a and b have no error bars. I would expect large errors, especially at the high voltages (>50 mV) considering the wide current variation of example traces in Figure 3a and histograms shown in the SI. The IV curves also have no associated n numbers in the figure legend or text, these must be included to show how many individual pore insertions the IV curves are representative of. The conductance error is also surprisingly low, considering the aforementioned current variation seen in Figure 3a and SI figures. Again, n numbers need to be included here.

Our response: As suggested, we have added error bars for Fig. 3a,b. These relative errors were obtained by dividing the width between the two half-height points by 2 (e.g., 10.0 pA / 2 = 5.0 pA in the case shown on the right).

Based on the Figure A and B presented above, the number of pore insertions is 3 for current traces at 60 mV, 1 for other voltages and 1 for I-V curves in a) and b). We have added this and other similar statements in the captions of Figures 3, S14 and S16.

As for “The conductance error is also surprisingly low, considering the aforementioned current variation seen in Figure 3a and SI figures”, this is because we used the mean current values obtained from their respective histograms (Figures S12, S13 and S15) to plot I-V curves. And these mean values roughly correspond to the membrane insertion of one single channel in most cases (though with large relative errors), and thus lead to a nearly linear I-V curve, yielding a conductance value with a low relative error.

Comment 5: Lastly, there is limited information on the nanopores anti-cancer activity. A graphical figure showing concentration dependent cell death in comparison to control cells and known anticancer drugs would greatly improve this section. Further experiments showing cell-binding with fluorescent tags would also improve the validity of this section. From the data it is not know whether the nanopores are disrupting the cell membrane, or are being internalized and disrupting intracellular membranes.

(1) Haris, P. I.; Molle, G.; Duclouhier, H. Conformational Changes in Alamethicin Associated with Substitution of Its α -Methylalanines with Leucines: A FTIR Spectroscopic Analysis and Correlation with Channel Kinetics. *Biophys. J.* 2004, 86 (1 1), 248–253. [https://doi.org/10.1016/S0006-3495\(04\)74100-4](https://doi.org/10.1016/S0006-3495(04)74100-4).

Our response: As suggested, we have carried out additional studies on normal cells (human normal liver LO2 cells and human embryonic kidney HEK 293T cells) and known anticancer drugs (doxorubicin, paclitaxel and cisplatin). Showing nanopores’ low cytotoxicities to normal cells and excellent selectivities to cancer cell, these results are discussed in Page 6 of the main text and summarized in Table 2, with graphical data presented in Figures S26-S28.

Together with other additional experiments to shed lights on the membrane integrity in the presence of nanopores and cell death via the apoptotic pathway, we have added the *italicized* discussions into Pages 6 and 7 of the main text.

Ch-C1 mediates potent and specific anticancer effects

..... Among them, the most potent **Ch-C1** exhibits an IC_{50} value of as low as $3.8 \mu M$, which is comparable to those of the well-known chemotherapeutic agents doxorubicin ($1.5 \mu M$) and paclitaxel ($8.2 \mu M$) and much lower than that of cisplatin ($> 500 \mu M$) (Table 2 and Figure S26). For human primary glioblastoma cell line (U87-MG) that are refractory to be treated, the above-mentioned three chemotherapeutic agents (cisplatin, doxorubicin and paclitaxel) show no activity ($IC_{50} > 500 \mu M$, Table 2 and Figure S27).....

These good anticancer activities of **Ch-C1** promoted us to further evaluate its selectivity and safety towards normal cells by performing the cell viability assay on human normal liver cells (LO2 cell line) and human embryonic kidney cells (HEK 293T cell line). Desirably, the determined IC_{50} values of **Ch-C1** against LO2 and 293T cells were 70.3 and $>500 \mu M$, respectively (Figure S28). Defined as the ratio of IC_{50} values between normal and cancer cells, the selectivity indexes (SI) determined for **Ch-C1** molecules are 18.5 and > 130 against LO2 and 293T cells, respectively. These high SI values are more than one order of magnitude higher than those of doxorubicin and paclitaxel⁸⁹⁻⁹⁰, indicating comparably lower cytotoxicities of **Ch-C1** to normal cells than doxorubicin and paclitaxel. We then employed mice blood cells to test the *in vitro* hemolytic activity of **Ch-C1**. To our delight, **Ch-C1** also shows a low hemolytic activity. The concentration causing 50% hemolysis of red blood cells (HC_{50}) is $127.3 \mu M$ (Figure S29), corresponding to a high SI value of 38.5. This high specificity may be caused by the high level expression of cholic acid receptors on hepatocarcinoma cells that leads to higher uptake of cholic acid-containing channel molecules⁹¹⁻⁹².

Two types of dye molecules (PI and DAPI) were applied to assess the membrane integrity in the presence of **Ch-C1** molecules. While DAPI molecules can enter the cells at high concentrations and become blue upon binding to the AT regions of dsDNA, membrane-impermeable PI enters cells having compromised membranes and binds tightly to the intracellular nucleic acids to emit red fluorescence when excited at 535 nm. After the incubation of HepG2 cells with various concentrations of **Ch-C1** (0, 1, 4 and $16 \mu M$) for 36 h, the cells were fixed, stained with PI and DAPI, and analyzed under a laser confocal microscope. As can be seen from Figure 4, the DAPI-stained cells reveal intact cell membrane, confirming that pore-forming **Ch-C1** does not disrupt the cell membranes to a noticeable extent at concentrations of up to $16 \mu M$. And the PI-stained cells do suggest that **Ch-C1** make the membrane leaky and more permeable via forming wide pores.

Figure 4. Cell imaging of HepG2 cells treated with **Ch-C1** at concentrations of $1 \mu M$, $4 \mu M$ and $16 \mu M$ for 36 h, followed by staining with blue DAPI and red PI dyes. Blue and red images were merged by Image J. DAPI = 4',6-diamidino-2-phenylindole; PI = propidium iodide.

It has been shown that disruption of cellular homeostasis caused by leaky cell membrane leads to cell apoptosis via the caspase signaling pathway⁹³⁻⁹⁵. Applying dead cell apoptosis kit, we treated the HepG2 cells with **Ch-C1** at 16 μM for 24 h, stained the cells using both green Annexin V- FITC conjugate and red PI dyes and sorted the cells by flow cytometry. Considering that cells stainable by Annexin V-FITC conjugate correspond to early or later apoptotic cells (Figure 5a), the fact that the percentage of apoptotic cells substantially increases from 1.07% to 8.88% with increased concentrations of **Ch-C1** from 0 to 16 μM establishes the capability of **Ch-C1** to induce cell apoptosis. Some characteristic proteins involved in apoptosis include (1) apoptosis initiator protein (caspase-9) that initiates cell killing but is cleaved during early apoptosis^{96,97}, (2) poly(ADP-ribose) polymerase (PARP) cleaved by the activated caspase 9 to facilitate apoptosis by preventing DNA repair and (3) anti-apoptotic protein (Bcl-2) that undergoes apoptosis- induced inhibition. Therefore, following an apoptotic stimulus, decreased presences of caspase 9, PARP and Bcl-2, together with the increased presence of cleaved PARP, are expected. And these are indeed what we observed when we treated HepG2 cells with **Ch-C1** at 16 μM for 6 h (Figure 5b,c), prompting us to conclude that **Ch-C1**-enhanced membrane permeability can induce HepG2 cell apoptosis via the caspase 9 pathway.

Figure 5. (a) Evaluation of HepG2 apoptosis by flow cytometry, with **Ch-C1** at 0 μM and 16 μM for 24 h and cells stained using green Annexin V-FITC conjugate and red PI dye. Annexin V = Intracellular protein of the annexin family that recognizes phosphatidylserines; FITC = Fluorescein isothiocyanate. (b) Immunoblot assay for Caspase 9, PARP and Bcl-2 in HepG2 cells treated with up to 16 μM of **Ch-C1** for 24 h. Results were analyzed via Image J and reported as histograms by graphpad prism 8.01 in (c). Symbols * and ** stand for significant differences between the control group and other groups, with $P < 0.05$ and 0.01 , respectively.

With respect to “Further experiments showing cell-binding with fluorescent tags would also improve the validity of this section”, we have synthesized **Ch-C1** conjugated to a pyrene molecule (**pyrene-Ch-C1**) conjugate. Nevertheless, likely due to the formation of a self-quenching dimer (**pyrene-Ch-C1**)₂ as inferred from the existence of dimeric (**Ch-C1**)₂ in the HR-MS (Figures S21 and S22), **pyrene-Ch-C1** is not fluorescent after membrane insertion. Although we can’t use the fluorescent tag to check “whether the nanopores are disrupting the cell membrane, or are being internalized and disrupting intracellular membranes”, we do believe that channels such as **Ch-C1** induce cell death primarily by permeabilizing the outer membrane. There are two-fold reasons. (1) **Ch-C1** has two lipid anchoring groups that structurally resemble the hydrophobic cholesterol group that, when combined with the poor water solubility of **Ch-C1**, should prevent **Ch-C1** from leaving the outer membrane for the intracellular membrane. (2) Given that cellular internalization often requires recognition by the cell surface receptor, cholesterol-like **Ch-C1** likely can’t be efficiently recognized by the surface receptors, thereby making internalization less likely. That is, residing in outer membrane should dominate over internalization into intracellular membrane.

REVIEWER COMMENTS

Reviewer #1 (Remarks to the Author):

Overall, this manuscript is much improved. The results of the additional cell experiments may have insight into the physiological mechanism of this artificial pore although I am not an expert on cell biology. The results of the cell experiments may be convincing, but the experimental details on the additional experiments are needed. Please add them including how did you take the control experiments (DMSO concentration etc..) and the detail of the statical methods.

I agree that the cholesterol should associate with the Ch molecules from the results of the NMR titration. However, I am still concerned about the molecular basis of the pore-formation. The authors state the formed pore as a “barrel-stave pore”. In the several reports about protein (10.1021/acs.analchem.7b01550) and peptide (10.1021/acsabm.8b00835 and 10.1021/acsomega.9b01033) pore-formation based on the electrophysiological measurements, the barrel-stave pore is most likely to show a step signal. The shape of your presented current signals seems to be the multilevel or erratic signal that are assigned into the toroidal or random disruption model of peptide pores. Can you add a more precise discussion on the molecular basis of the pore-formation in terms of the previous reports?

Reviewer #2 (Remarks to the Author):

Overall, the comments and additional experiments have answered the concerns to a satisfactory level. However, there are two final comments that should be addressed before publication of this manuscript.

Firstly, the reviewer rebuttal mentions that “the number of pore insertions is 3 for current traces at 60 mV, 1 for other voltages and 1 for I-V curves in a) and b)”. Some major conclusions are drawn with regards to pore insertion in experiments where $n = 1$ – with no repeat measurement at the majority of voltages. It is typical to obtain at least an $n = 3$ for each voltage and take an average (with many researchers opting to obtain tens or hundreds of insertions for a more accurate characterisation of pore activity), especially with regards to an irregular pore that by your own admission likely produces “a mixture of nanopores, having different diameters or different channel over cholesterol molar ratios.”. $n = 1$ for all voltages except 60 mV is not sufficient to draw conclusions and publication of this data.

Secondly, the authors have added a more detailed experimental methodology for determining the anticancer effects of Ch-C1 in comparison to known anticancer agents. My only concern with this new data are the cell lines used in the viability assays of “normal” human cells. Firstly, L02 liver cell lines

should not be used as a model for liver cells as they are a HeLa contaminated cell line and thus not originating from liver but from cervix (https://web.expasy.org/cellosaurus/CVCL_6926). Not only that, but HeLa cells are also a cancerous cell line, which may impact the hypothesis that Ch-C1 acts as an anticancer agent. In addition to this, HEK293 cells are also far from a normal human cell line being immortalised and also tumorigenic, containing an abnormal chromosome number of 64. Whilst not deriving from a cancerous cell line, there may be other more suitable cell lines to use as a control for toxicity of a normal human cell.

Comments by Reviewer 1 and Our Responses

Overall, this manuscript is much improved. The results of the additional cell experiments may have insight into the physiological mechanism of this artificial pore although I am not an expert on cell biology. The results of the cell experiments may be convincing, but the experimental details on the additional experiments are needed. Please add them including how did you take the control experiments (DMSO concentration etc..) and the detail of the statical methods.

Our response: Experimental details on additional experiments including *in vitro* anticancer study via MTT assay, cell membrane integrity assessment, flow cytometry assay and immunoblot analysis are provided in the “**method**” section and supporting information. In a typical experiment, 0.5% DMSO solution served as a negative control. In a MTT assay, cell viabilities with the addition of channel molecules at various concentrations were calculated via the following equation. The cell viabilities versus logarithm of channel concentrations were plotted and the IC₅₀ values were calculated using a nonlinear regression curve fit with Graphpad Prism 8.0.1.

$$\%Cell\ Viability = \frac{OD_{490}(\text{channel})}{OD_{490}(0.5\% \text{ DMSO})} \times 100\%$$

I agree that the cholesterol should associate with the Ch molecules from the results of the NMR titration. However, I am still concerned about the molecular basis of the pore-formation. The authors state the formed pore as a “barrel-stave pore”. In the several reports about protein (10.1021/acs.analchem.7b01550) and peptide (10.1021/acsabm.8b00835 and 10.1021/acsomega.9b01033) pore-formation based on the electrophysiological measurements, the barrel-stave pore is most likely to show a step signal. The shape of your presented current signals seems to be the multilevel or erratic signal that are assigned into the toroidal or random disruption model of peptide pores. Can you add a more precise discussion on the molecular basis of the pore-formation in terms of the previous reports?

Our response: Thank you for this very constructive comment. We have carefully read all three references recommended by the Reviewer, and have cited two of them as Refs 84 and 85 in our revised paper. Yes, we totally agreed with the Reviewer in that the pores formed by our molecules should be better assigned as the toroidal pores since most of the single channel current traces show multiple level signals as illustrated by the following representative recordings from - 100 mV to 100 mV presented in Figure 3a (For more current traces, see Figures S17 and S19).

Accordingly, in addition to replacing all “barrel-stave pore” with “toroidal pore” throughout the main text, we have added the following discussions **in red**:

Added into the revised manuscript: *At other voltages, many multiple level transitions were observed in the current traces, suggesting the existence of toroidal pores of*

varied diameters^{84,85}. These varied diameters result from dynamic breathing-type interactions between **Ch-C1** and **Ch** and these interactions might be further influenced by the lipid molecules that are also in constant movement.

Figure 3. (a) Single channel current traces of **Ch-C1** recorded in symmetric (*cis* chamber = *trans* chamber = 1 M KCl) baths, the red lines refer to the mean current values for plotting current-voltage (I-V) curve for obtaining the ion conductance (γ) and pore size (1.67 nm) for **Ch-C1**.

Comments by Reviewer 2 and Our Responses

Overall, the comments and additional experiments have answered the concerns to a satisfactory level. However, there are two final comments that should be addressed before publication of this manuscript.

Firstly, the reviewer rebuttal mentions that “the number of pore insertions is 3 for current traces at 60 mV, 1 for other voltages and 1 for I-V curves in a) and b)”. Some major conclusions are drawn with regards to pore insertion in experiments where $n = 1$ – with no repeat measurement at the majority of voltages. It is typical to obtain at least an $n = 3$ for each voltage and take an average (with many researchers opting to obtain tens or hundreds of insertions for a more accurate characterisation of pore activity), especially with regards to an irregular pore that by your own admission likely produces “a mixture of nanopores, having different diameters or different channel over cholesterol molar ratios.”. $n = 1$ for all voltages except 60 mV is not sufficient to draw conclusions and publication of this data.

Our response: As suggested, we have carried out single channel current trace measurement from -100 mV to 100 mV for **Ch-C1** and **Ch-C4** in triplicate run. From the linear *I-V* curves, the single-channel conductance values for **Ch-C1** in triplicate run were determined to be 534.7 ± 18.1 pS, 523.6 ± 11.9 pS and 430.9 ± 11.9 pS (Figures S13-S19). The average conductance value for **Ch-C1** was 496.4 ± 46.5 pS, giving rise to an average pore size of 1.61 ± 0.09 nm. One representative run ($\gamma = 534.7 \pm 18.1$ pS and pore size = 1.67 nm) was illustrated in Figure 3a.

Figure 3. (a) Single channel current traces of **Ch-C1** recorded in symmetric (*cis* chamber = *trans* chamber = 1 M KCl) baths, the red lines refer to the mean current values for plotting current-voltage (*I-V*) curve for obtaining the ion conductance (γ) and pore size (1.67 nm) for **Ch-C1**.

Similarly, the single-channel conductance values for **Ch-C4** in triplicate run were determined to be 175.7 ± 3.1 pS, 190.7 ± 3.6 pS and 199.0 ± 7.1 pS (Figures S20-27),

giving rise to an average conductance value of 188.5 ± 9.6 pS and an average pore size of 0.94 ± 0.03 nm for **Ch-C4**.

Secondly, the authors have added a more detailed experimental methodology for determining the anticancer effects of Ch-C1 in comparison to known anticancer agents. My only concern with this new data are the cell lines used in the viability assays of “normal” human cells. Firstly, L02 liver cell lines should not be used as a model for liver cells as they are a HeLa contaminated cell line and thus not originating from liver but from cervix (https://web.expasy.org/cellosaurus/CVCL_6926). Not only that, but HeLa cells are also a cancerous cell line, which may impact the hypothesis that Ch-C1 acts as an anticancer agent. In addition to this, HEK293 cells are also far from a normal human cell line being immortalised and also tumorigenic, containing an abnormal chromosome number of 64. Whilst not deriving from a cancerous cell line, there may be other more suitable cell lines to use as a control for toxicity of a normal human cell.

Our response: We thank the reviewer for these highly insightful comments. As suggested, we have performed the cell viability assays on human liver epithelial cells (THLE-2) and human renal proximal tubular epithelial cells (HK-2) as a control for toxicity of normal human cells. The determined IC_{50} values for **Ch-C1** against THLE-2 cells and HK-2 cells were 47.5 and >500 μ M, respectively. The corresponding selectivity indexes (SI) were 12.5 and > 130, respectively, indicative of the potent and specific anticancer activity of **Ch-C1**. These new data have been added into the revised manuscript.

REVIEWERS' COMMENTS

Reviewer #1 (Remarks to the Author):

My comments were addressed in a satisfactory manner. I am okay with the paper being published as is.

Reviewer #2 (Remarks to the Author):

The points raised in the second round of reviewing have been successfully addressed. The manuscript is now publishable in Nature Communications.

Comments by Reviewer 1 and Our Responses

The results of the additional cell My comments were addressed in a satisfactory manner. I am okay with the paper being published as is.

Our response: We thank reviewer 1 for endorsing publication of our manuscript in Nat. Commun..

Comments by Reviewer 2 and Our Responses

The points raised in the second round of reviewing have been successfully addressed. The manuscript is now publishable in Nature Communications.

Our response: We thank reviewer 1 for endorsing publication of our manuscript in Nat. Commun..